# Psychotherapy Incorporating Equine Interaction as a Complementary Therapeutic Intervention for Young Adults in a Residential Treatment Program during the COVID-19 Pandemic

Katie Holtcamp [1], Molly C. Nicodemus [2,*], Tommy Phillips [3], David Christiansen [4], Brian J. Rude [2], Peter L. Ryan [5] and Karen Galarneau [6]

1. Counseling Services, Dogwood Equine Connection Therapy Center, Starkville, MS 39760, USA
2. Animal & Dairy Sciences Department, Mississippi State University, Starkville, MS 39762, USA
3. School of Human Sciences, Mississippi State University, 255 Tracy Drive, Starkville, MS 39762, USA
4. Large Animal Medicine Department, College of Veterinary Medicine, Mississippi State University, Starkville, MS 39762, USA
5. Office of Provost and Executive Vice President, Mississippi State University, Starkville, MS 39762, USA
6. Counseling Services, College of Veterinary Medicine, Mississippi State University, Starkville, MS 39762, USA
* Correspondence: mnicodemus@ads.msstate.edu

**Abstract:** Substance use disorder has become an epidemic in the young adult population across the United States, and these numbers rose during the COVID-19 pandemic. Psychotherapy incorporating equine interaction has emerged to show promise in the mental health community as a complementary form of therapy for this age group and offered a viable treatment option during the pandemic due to the outdoor nature of the treatment environment. However, research concerning its use within a residential treatment program was lacking. The objective of this study was to evaluate the efficacy of psychotherapy incorporating equine interaction in a residential treatment program during the COVID-19 pandemic for developing an emotionally safe environment for learning for young adults. Participants (ages 18–25 years) were those in a substance abuse residential treatment program utilizing psychotherapy incorporating equine interaction during the COVID-19 pandemic. Participants were involved in weekly equine therapy for 2–7 weeks. Participants were divided according to length of stay at the residential facility and participation level with equine interactive activities. Assessment of emotional safety and long-term memory development was performed at the beginning and end of the treatment program. The development of memories centered around equine information that was covered during the treatment program. Semantic memory was assessed using a self-reporting knowledge exam and procedural memory was assessed using a skill evaluation. Emotional safety was determined using a self-reporting survey instrument. Paired *t*-tests determined significant improvement in emotional safety ($p = 0.02$) and semantic ($p = 0.01$) and procedural ($p = 0.00$) memory for all participants by the end of the program. The one-way analysis of variance indicated length of stay and participation level were not significant indicators of emotional safety (length of stay: $p = 0.91$, participation level: $p = 0.98$) and semantic (length of stay: $p = 0.09$, participation level: $p = 0.60$) and procedural (length of stay: $p = 0.25$, participation level: $p = 0.09$) memory development. These results suggest psychotherapy incorporating equine interaction was an efficient complementary therapeutic intervention for developing emotional safety and encouraging learning in a young-adult residential addiction treatment program during the COVID-19 pandemic.

**Keywords:** psychotherapy incorporating equine interaction; substance abuse disorder; residential treatment program; young adult

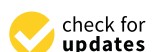



## 1. Introduction

Substance use disorder (SUD) is an epidemic in the United States [1,2]. In 2016, research indicated 20.3 million individuals aged 12 and older experienced SUD [3], and with

the coronavirus disease 2019 (COVID-19) evolving into a pandemic [4,5], these numbers were further exacerbated [6,7]. This rise in numbers associated with substance abuse is concerning, as opioids are the leading cause of death in adults 50 years and younger in the United States [8]. With SUD, the individual goes through recurrent patterns of use of alcohol and/or illicit drugs that cause significant diminishment in healthy functioning, which impacts social, physical, and mental well-being [3], and the majority of people are unable to break this cycle, falling into relapse after treatment [9].

## 1.1. Young Adult Population

While SUD affects all age groups within the United States population, the most common population battling SUD is that found on college campuses [10]. In 2016, 34.8%, almost 12 million, of surveyed college-age adults ages 18–25 years old were using an illegal substance on a regular basis [3]. In 2020, with the introduction of the COVID-19 pandemic [4,5], mental health challenges including loneliness, depression, and anxiety plagued college campuses [11], which in turn only compounded the abuse of substances [6]. While the abuse may not always be deadly, the impact can be negative on students [12]. On the mild end of the spectrum, 46% of students surveyed claimed to have done something embarrassing when under the influence of a substance, while 44% did not complete a school assignment [12]. For others, the consequences were more severe, as 27% reported needing more drugs to obtain the desired effect, while 15% used more substances to avoid withdrawal symptoms [12]. Furthermore, these numbers do not address the potential long-term deficit in cognitive functioning that can result from continued abuse of substances [13], and these deficits can be detrimental to the college student [10]. Even for those that turn to interventions to avoid these consequences, the relapse rates can be discouraging. Young adults in recovery are reported to have a relapse rate of 60% or higher due to the nature of life events at that age [14]. To manage these risks, experts focus on shorter and more engaged physically interactive treatment methods, such as those found in residential treatment programs, as these programs have been proven more effective with lower relapse rates for young adults than those reported for traditional out-patient weekly talk therapy [15]. These traditional treatment options, however, faced further challenges beyond relapse rates with the onset of the COVID-19 pandemic, as quarantine guidelines limited direct interaction with therapists [4,6]. As such, telehealth options were explored, but limitations associated with technology prevented widespread use [7].

## 1.2. Psychotherapy Incorporating Equine Interaction

With the COVID-19 pandemic and the associated rise in mental health disorders came challenges in therapeutic approaches, both in addressing the obstacles associated with social distancing guidelines and in meeting the increasing demand in the need for accessible treatment [11,16,17]. One non-traditional therapy option found within residential programs that provided an environment to meet these challenges associated with the COVID-19 pandemic was psychotherapy incorporating equine interaction (PIE), due to the outdoor nature of the therapeutic intervention [16]. PIE is a form of equine assisted service (EAS) that moves the therapy outside of the traditional therapy environment and into the equine environment [18]. Along with meeting social distancing guidelines during the pandemic [5,6], this environmental change proved to be promising for patients experiencing mental health issues [2,4]. The goal of this form of therapy is to create an emotionally safe environment that promotes learning [19,20]. Horses are an ideal tool for this therapy process as they are instinctively social animals living in herds, so that when a human spends time in their presence, they become a part of that natural social dynamic found in the herd [21]. This engagement happens without the individual having to know anything about the horse or its behavior, naturally creating a disarmament of the conscious mind so that the subconscious can work [20]. This coupled with a successful history of retention rates has made PIE a viable complementary approach to traditional treatment programs [18].

*1.3. Present Study*

Although anecdotal evidence points to patient improvement [18,20], research is limited when evaluating the effectiveness of PIE specific to residential addiction treatment programs for young adults, with data only reporting positive retention rates [22]. Further, research concerning the impact of this therapeutic intervention on SUD residential treatment patients implemented during the COVID-19 pandemic is lacking. Therefore, the objective of this study was to evaluate the effectiveness of PIE in creating an emotionally safe environment for learning for young adults in a residential addiction treatment program during the COVID-19 pandemic. The study hypothesized that the acquisition of equine-related long-term memory, specifically semantic and procedural memory, would occur with the development of emotional safety for young adults participating in PIE sessions at a residential addiction treatment program during the COVID-19 pandemic, and due to the efficacy of this therapeutic intervention, stay within the facility, and the offering of equine therapeutic-related activities associated with PIE could be minimized. To the best of our knowledge, this is the first study to examine the impact of equine interaction on emotional safety and learning within young adults with SUD. Further, this is the first to explore the use of psychotherapy incorporating equine interaction within a residential addiction treatment program during the COVID-19 pandemic. Although the mental health community has transitioned into the post-COVID-19 era [4], the understanding of the efficacy of this novel therapeutic intervention can be of value to mental health professionals as they evaluate treatment options for the young adult population with SUD while considering potential compounding circumstances during the treatment process, as observed during the COVID-19 pandemic [4–6].

## 2. Materials and Methods

*2.1. Recruitment of Human Participants*

Individuals were recruited from the young adult program at the American Addiction Centers' Oxford Treatment Resolutions Center in Oxford, Mississippi in 2020 during the first year of the COVID-19 pandemic. Recruitment for this study was performed in person at the time of entry of the individual into the center utilizing the staff at the center. The young adult program consisted of individuals between 18 and 25 years of age. All individuals in this program were given the opportunity to participate in weekly PIE sessions as a part of their treatment program for substance abuse. Individuals that opted to participate in the PIE sessions did not have to participate in this study, as the study was voluntary and was not required for the PIE treatment program.

Participants for this study were individuals actively undergoing drug and/or alcohol treatment at the residential treatment center that included weekly PIE sessions. In addition to a once-a-week PIE session, participants could freely select participation level in additional equine interactive activities during their stay at the center and could determine their level of involvement with the study. A lack of involvement in any of these equine interactive activities did not disqualify the individual from the study. All aspects of the study, including recruitment, assessments, and equine interactive activities were evaluated and approved by the Institutional Review Board at Mississippi State University prior to being available to participants. All equine interactive activities as they pertained to the welfare of the equine were evaluated and approved by the Institutional Animal Care and Use Committee at Mississippi State University prior to the treatment process.

*2.2. Participant Therapy Activities*

Participation in equine interactive activities was recorded weekly for each participant by the mental health professionals coordinating the PIE sessions at the treatment center. The horses used for these equine interactive activities were property of Oxford Treatment Center and were housed at the residential facility, allowing for the opportunity of participant interaction with these horses on a daily basis. Thus, outside of the primary PIE curriculum, participants had the opportunity to take part in various other equine interactive activities.

During the participant's stay at the residential facility, health status was monitored by the physicians of the treatment center. The mental health professionals conducting the PIE sessions were notified if physicians at the facility encountered any health-related concerns with the study participants, including contraction of COVID-19 by participants. Health monitoring followed guidelines outlined by the Joint Commission [23] along with that given by the Mississippi Department of Mental Health [24]. The facility maintained during this study Behavioral Health Care & Human Services Accreditation through the Joint Commission, and as such, accreditation guidelines were met concerning the health-care of patients at the facility, including compliance concerning health status reporting procedures [23]. Reporting included any confirmed cases of COVID-19. Following accreditation guidelines [23] and state mental health policies [24], physicians at the facility were responsible for making the final decision of continued participation in any aspect of the equine interactive activities, including the weekly PIE sessions.

Participation in the once-a-week PIE sessions ranged between 2 and 7 weeks, depending on the participant's stay at the residential center. Length of stay (LOS) of the participant at the residential center was dependent on the individual's schedule, their mental health-care needs, and their financial abilities, including insurance coverage. Participation in the study had no impact on the participants' LOS. Participants were divided into groups according to their LOS at the residential center: 2–4 weeks LOS or 5–7 weeks LOS.

### 2.2.1. Primary Curriculum

All participants attended a weekly PIE session that rotated through the following topics: separation anxiety, family dynamics, communication, trust and patience, boundaries, external triggers, and powerlessness. This curriculum was modeled similarly to the OK Corral Series curriculum created by Greg Kersten [25] and was the curriculum used at the center prior to the COVID-19 pandemic. No changes were made to the curriculum for this study other than the use of masks for covering mouth and nostrils by mental health professionals, facility staff, and patients. The length for each weekly PIE session for this project was an hour and a half, consisting of group discussion and hands-on activities both on and off of the horse. Participant attendance and activities related to these sessions were documented by the mental health professionals conducting the PIE sessions at the treatment center.

### 2.2.2. Additional Equine Interaction

All participants had an opportunity to participate in additional equine interactive activities, which included attending extra PIE sessions; contributing to daily barn duty activities, including feeding and grooming horses and cleaning stalls; and taking part in trail rides. Participation in these activities for each study participant was documented and was voluntary. All activities were coordinated and monitored by the mental health professionals associated with the PIE sessions at the treatment center. Additional equine interaction ranged in time between one and two hours per session above the assigned weekly PIE session. Participants were divided into the number of sessions, including various equine interactive activities beyond the weekly PIE sessions, that an individual participated in. The participation level (PL) according to the number of weekly sessions participated in was grouped in the following manner: 1–2 weekly equine sessions, 3–4 weekly equine sessions, or 5–7 weekly equine sessions. PL included the one assigned weekly PIE session that was a part of the primary PIE curriculum for the treatment center. Grouping of PL was further divided by the LOS (2–4 weeks or 5–7 weeks), forming the following six groups for this study: Group A—participants with 2–4 weeks LOS and PL of 1–2 equine sessions; Group B—participants with 2–4 weeks LOS and PL of 3–4 equine sessions; Group C—participants with 2–4 weeks LOS and PL of 5–7 equine sessions; Group D—participants with 5–7 weeks LOS and PL of 1–2 equine sessions; Group E—participants with 5–7 weeks LOS and PL of 3–4 equine sessions; and Group F—participants with 5–7 weeks LOS and PL of 5–7 equine sessions. Groups A and D had the lowest PL with 1–2 equine sessions,

followed by Groups B and E with 3–4 equine sessions, while Groups C and F had the most PL with 5–7 equine sessions.

### 2.3. Emotional Safety Evaluation

All participants were given a self-reporting emotional safety survey instrument at the beginning and end of their residential treatment program. The survey instrument consisted of 60 questions divided into the four categories of emotional safety (personal security, connectivity, self-esteem, and respect) [19]. Questions for each category of emotional safety (15 questions for each category) came directly from previously validated assessments that included the following: Personal Security-GAD-7 [26] and Emotional Needs Scale [27]; Connectivity-Social Connectedness Scale [28]; Self-Esteem- Self-Esteem Inventory [29]; and Respect- Trust/Respect Assessment [30]. When scoring this survey, questions with a positive point of view were scored as Always—1, Sometimes—2, Seldom—3, Never—4, and N/A—0. An example of this type of question included "I have self-worth". Questions with a negative perspective were scored in an opposite manner: Never—1, Seldom—2, Sometimes—3, Always—4, and N/A—0. An example of this type of question included "I have trouble staying motivated to complete a task". The total scores were evaluated in their relation to the ideal total score of 60 so that the ideal score for each category was a score of 15. A lowering of the emotional safety score closer to the ideal score was interpreted as an improvement in emotional safety.

### 2.4. Semantic Memory: Equine Knowledge Exam

To evaluate equine-related long-term memory development specific to semantic memory, all participants were given an equine knowledge exam at the beginning and end of their residential treatment program. The 22 questions making up the exam were taken directly from the Certified Horsemanship Association (CHA) Instructor Manual Level 1 [31]. Questions were divided up to focus on the four areas of horsemanship covered in the PIE curriculum: horse management/care, grooming and tacking procedures, riding activities, and basic equine behavior. This exam consisted of half of the questions formatted as multiple-choice questions and half formatted as true or false questions, with each question given 1 point for a correct answer. The maximum score for the exam was a 22, with no points given for incorrect or unanswered questions.

### 2.5. Procedural Memory: Equine Handling Skill Evaluation

To assess the development of equine-related long-term memory that was specific to procedural memory, all participants were evaluated within the equine environment by the mental health professionals conducting the PIE sessions as the participants performed various equine handling skills [32]. The skill evaluation was completed during the same week as the exam, with evaluation questions being the same for both the pre- and post evaluations. The evaluation consisted of 14 questions that were answered by the mental health professionals as they assessed their participants performing within the therapy session. Scoring utilized a 1 to 4 rating scale for assessing the level of proficiency and comfort with and around horses. Questions came from the CHA Instructor Manual Level 1, with questions divided equally into four assessment areas: abilities in equine care, quality in barn management, skills in horsemanship activities, and skills in team building during equine activities [31]. The maximum possible score for this evaluation was 56 points. If a participant did not complete an activity during the session, a score of zero was given for that assessment question, while an ideal completion of the activity following correct procedures as outlined in the CHA manual [31] was given a score of four.

### 2.6. Statistical Analysis

Descriptive statistics were determined for the emotional safety evaluation, equine knowledge exam, and equine handling skill evaluation. Descriptive statistics included means and standard deviations. A paired-samples *t*-test was performed using IBM SPSS

Statistics 26 (Armonk, NY, USA) to compare the pre- and post scores for the three evaluation methods. For further statistical analysis, participants were divided into six groups (Groups A–F) based on LOS and PL. For the emotional safety evaluation, equine knowledge exam, and equine handling skill evaluation, the differences between pre- and post scores were determined for each of the six groups and two-sample *t*-tests were performed between each group. In addition, multivariable linear regression models were utilized to determine if a relationship was present concerning emotional safety and equine knowledge exams and between emotional safety and equine handling skill evaluations. A one-way ANOVA with post hoc Bonferroni analysis was also conducted to define any significant variations in pre- and post scores between groups. Statistical significance was set at $p = 0.05$.

## 3. Results

### 3.1. Participation

128 individuals, 54 females and 74 males, participated in the initial (pre) evaluations ($n = 128$). For the final (post) evaluation, 61 individuals, 28 females and 33 males, completed evaluations prior to discharge ($n = 61$). The ages for the initial evaluations were $21.25 \pm 2.06$ years old, with ages in the final evaluations being $21.44 \pm 1.98$ years old. For all evaluations, the breakdown of participants per groups for the pre- and post scores were as follows: Group A (pre scores: $n = 62$, post scores: $n = 11$), Group B (pre scores: $n = 20$, post scores: $n = 11$), Group C (pre scores: $n = 5$, post scores: $n = 5$), Group D (pre scores: $n = 12$, post scores: $n = 12$), Group E (pre scores: $n = 21$, post scores: $n = 21$), and Group F (pre scores: $n = 8$, post scores: $n = 8$). This breakdown of participation per group was the same for all three evaluation methods used in this study. Groups C, D, E, and F had no drop in participation from pre- to post scores. Only 17.74% and 55% of the pre score participants in Groups A and B, respectively, completed the post score evaluations. A lack of participation in post score evaluations for those in Groups A and B was documented as associated with scheduling conflicts related to release dates from the treatment center, where the researchers were unable to follow up on final evaluations. No individual in Groups A and B that completed the initial evaluation refused participation in the final evaluation. Nevertheless, for the remainder of the study, pre score evaluations for Groups A and B only focused on those that completed the post score evaluations, and all statistical analyses were kept to those participating in both the pre- and post score evaluations. A lack of participation for all groups was not associated with any aspect of health-related issues linked to COVID-19, as reported by the physicians of the facility.

### 3.2. Emotional Safety Evaluation

When all groups were combined, a significant difference between pre- and post scores was found for the total emotional safety score ($p < 0.05$), with personal security and self-esteem demonstrating significant differences between pre- and post scores as post scores moved closer to the ideal score of 15 for those categories ($p < 0.05$, Table 1).

**Table 1.** Means (SD) for emotional safety evaluations for young adults participating in weekly psychotherapy incorporating equine interaction sessions with evaluations conducted at the beginning (pre) and end (post) of a residential treatment program.

| Categories of Emotional Safety | Pre Score ($n = 61$) | Post Score ($n = 61$) | *p*-Value |
|---|---|---|---|
| Personal Security | 39.98 (14.23) | 33.47 (6.87) | 0.001 |
| Respect | 27.86 (8.75) | 27.81 (5.53) | 0.967 |
| Self-Esteem | 26.00 (9.41) | 22.76 (5.87) | 0.017 |
| Connectivity | 32.63 (9.07) | 31.25 (6.04) | 0.277 |
| Total Evaluation Score | 126.47 (36.18) | 115.31 (18.96) | 0.022 |

Total scores lowering toward 60 indicated improvement in emotional safety, with ideal scores for each category of emotional safety set at 15.

When evaluating the individual groups, all groups except for Groups B and E demonstrated lowering of the total emotional safety scores by the end of the program, but the differences between pre- and post scores for total emotional safety and for the categories of emotional safety for all groups were not statistically significant ($p > 0.05$, Table 2). The one-way ANOVA indicated that LOS ($p = 0.91$) and PL ($p = 0.98$) were not significant indicators of emotional safety development, nor was the combination of LOS and PL a significant indicator of emotional safety ($p = 0.94$).

**Table 2.** Mean difference (SD) of emotional safety evaluation scores between pre- and post scores in regard to length of stay (LOS) and weekly participation level (PL) for young adults in a residential treatment program with weekly psychotherapy incorporating equine interaction sessions.

| Group Category | Personal Security | Respect | Self-Esteem | Connectivity | Total Emotional Safety |
|---|---|---|---|---|---|
| Group A ($n = 11$) | −5.82 (9.62) | −2.27 (5.66) | −3.18 (3.16) | −2.27 (6.07) | −13.55 (18.07) |
| Group B ($n = 4$) | 6.75 (17.75) | 4.50 (18.28) | 9.00 (10.99) | 9.00 (16.19) | 29.25 (61.26) |
| Group C ($n = 5$) | −5.00 (7.52) | −2.60 (3.91) | −6.40 (0.55) | 0.00 (2.28) | −14.00 (5.91) |
| Group D ($n = 12$) | −7.75 (4.07) | −5.08 (6.88) | −3.83 (3.67) | −4.50 (7.05) | −21.77 (14.97) |
| Group E ($n = 21$) | −9.95 (13.82) | 0.50 (8.98) | −6.75 (10.28) | −1.05 (7.73) | 29.25 (61.26) |
| Group F ($n = 8$) | −13.29 (8.40) | −1.29 (4.75) | −4.43 (8.99) | −3.00 (7.00) | −22.00 (24.90) |

Negative values indicated a lowering of scores by final evaluations. Groups were divided according to LOS and PL: Group A (LOS—2–4 weeks; PL—1–2 equine activities/week), Group B (LOS—2–4 weeks; PL—3–4 equine activities/week), Group C (LOS—2–4 weeks; PL—5–7 equine activities/week), Group D (LOS—5–7 weeks; PL—1–2 equine activities/week), Group E (LOS—5–7 weeks; PL—3–4 equine activities/week), Group F (LOS—5–7 weeks; PL—5–7 equine activities/week). No significant differences were found between groups ($p = 0.05$).

*3.3. Semantic Memory: Equine Knowledge Exam*

In evaluating all participants for the development of semantic memory associated with equine-related knowledge, while a significant difference was seen in overall scores when comparing pre- to post equine knowledge exam scores ($p < 0.05$, Table 3), the only categories for the exam that demonstrated significant differences between pre- and post scores were the riding and behavior categories ($p < 0.05$).

**Table 3.** Means (SD) for equine knowledge exam scores for young adults participating in weekly psychotherapy incorporating equine interaction sessions, with evaluations conducted at the beginning (pre) and end (post) of a residential treatment program.

| Categories of Equine Knowledge Exam | Pre Scores ($n = 61$) | Post Scores ($n = 61$) | *p*-Value |
|---|---|---|---|
| Grooming/Tacking | 3.27 (1.10) | 3.61 (1.16) | 0.15 |
| Riding | 6.14 (1.60) | 6.78 (1.49) | 0.02 |
| Equine Behavior | 2.56 (0.90) | 2.81 (0.75) | 0.05 |
| Equine Care | 4.05 (0.88) | 4.15 (0.85) | 0.49 |
| Total Exam Score | 16.02 (2.98) | 17.36 (2.64) | 0.01 |

The equine knowledge exam was scored on a 0–22 point scale.

As for differences between groups, while Groups A and C had similar LOS values, the two groups demonstrated significant differences between total exam scores when comparing the two groups ($p < 0.05$, Table 4). Similarly, while Groups C and F had the same

PL, significant differences were seen in total scores between the two groups ($p < 0.05$). The one-way ANOVA signified independently that LOS ($p = 0.09$) and PL ($p = 0.60$) were not significant indicators of equine knowledge attainment, nor were LOS and PL when the two were combined ($p = 0.62$).

**Table 4.** Mean difference (SD) of equine knowledge exam scores between pre- and post scores in regard to length of stay (LOS) and weekly participation level (PL) for young adults in a residential treatment program with weekly psychotherapy incorporating equine interaction sessions.

| Group Category | Grooming/Tacking | Riding Skills | Equine Behavior | Equine Care | Total Equine Knowledge |
|---|---|---|---|---|---|
| Group A (*n* = 11) | 0.55 (1.04) | 1.64 (2.42) | 0.64 (0.92) | 0.09 (0.70) | 2.91 [a] (3.02) |
| Group B (*n* = 4) | 0.50 (1.00) | 2.50 (1.71) | 0.50 (1.29) | 0.50 (1.00) | 1.75 (1.50) |
| Group C (*n* = 5) | −0.20 (0.84) | 0.00 (0.71) | 0.60 (0.89) | 0.20 (0.45) | 0.60 [a,b] (1.52) |
| Group D (*n* = 12) | −0.25 (1.36) | −0.08 (1.44) | 0.25 (0.75) | 0.25 (1.06) | 0.17 (1.85) |
| Group E (*n* = 21) | 0.00 (0.86) | 0.45 (1.57) | 0.00 (0.86) | 0.05 (0.89) | 0.40 (2.14) |
| Group F (*n* = 8) | 0.43 (2.07) | 0.86 (1.07) | 0.29 (0.76) | 0.00 (1.16) | 1.57 [b] (3.26) |

Negative values indicate a lowering of scores by final evaluations. Groups were divided according to LOS and PL: Group A (LOS—2–4 weeks; PL—1–2 equine activities/week), Group B (LOS—2–4 weeks; PL—3–4 equine activities/week), Group C (LOS—2–4 weeks; PL—5–7 equine activities/week), Group D (LOS—5–7 weeks; PL—1–2 equine activities/week), Group E (LOS—5–7 weeks; PL—3–4 equine activities/week), Group F (LOS—5–7 weeks; PL—5–7 equine activities/week). Superscripts indicate the following: [a] indicates a significant difference between Group A and Group C in Total Equine Knowledge and [b] indicates a significant difference between Group C and Group F in Total Equine Knowledge ($p = 0.05$).

When all participants were evaluated utilizing regression analysis, no correlation was found between emotional safety scores and equine knowledge exam scores (R = 0.02, $R^2$ = 0.00). When evaluating each group, the regression analysis determined only a weak positive correlation for Group B between emotional safety scores and equine knowledge exam scores (Table 5).

**Table 5.** Regression results for equine knowledge exams and equine handling skill evaluations in relation to emotional safety evaluations for young adults participating in a residential treatment program with weekly psychotherapy incorporating equine interaction sessions, separated by length of stay (LOS) and weekly participation levels (PLs).

| | Equine Knowledge Exam | | Equine Handling Skill Evaluation | |
|---|---|---|---|---|
| **Group Category** | **R** | **$R^2$** | **R** | **$R^2$** |
| Group A (*n* = 11) | 0.16 | 0.03 | 0.46 | 0.22 |
| Group B (*n* = 4) | 0.29 | 0.09 | 0.53 | 0.28 |
| Group C (*n* = 5) | 0.23 | 0.05 | 0.93 | 0.86 |
| Group D (*n* = 12) | 0.07 | 0.00 | 0.09 | 0.00 |

**Table 5.** *Cont.*

| | Equine Knowledge Exam | | Equine Handling Skill Evaluation | |
|---|---|---|---|---|
| **Group Category** | **R** | **$R^2$** | **R** | **$R^2$** |
| Group E ($n = 21$) | 0.19 | 0.04 | 0.19 | 0.04 |
| Group F ($n = 8$) | 0.22 | 0.05 | 0.07 | 0.00 |

Groups were divided according to LOS and PL: Group A (LOS—2–4 weeks; PL—1–2 equine activities/week), Group B (LOS—2–4 weeks; PL—3–4 equine activities/week), Group C (LOS—2–4 weeks; PL—5–7 equine activities/week), Group D (LOS—5–7 weeks; PL—1–2 equine activities/week), Group E (LOS—5–7 weeks; PL—3–4 equine activities/week), Group F (LOS—5–7 weeks; PL—5–7 equine activities/week).

*3.4. Procedural Memory: Equine Handling Skill Evaluation*

In evaluating all participants for the development of procedural memory associated with equine-related activities, the overall equine handling skill evaluation scores demonstrated a significant difference by the post evaluation ($p < 0.05$, Table 6). In addition, all four categories of the skill evaluation showed significant differences between pre- and post evaluation scores ($p < 0.05$).

**Table 6.** Means (SD) for equine handling skill evaluations for young adults participating in weekly psychotherapy incorporating equine interaction sessions, with evaluations conducted at the beginning (pre) and end (post) of a residential treatment program.

| Categories of Equine Handling Skill Evaluation | Pre Scores ($n = 61$) | Post Scores ($n = 61$) | *p*-Value |
|---|---|---|---|
| Abilities in Equine Care | 9.13 (4.00) | 13.89 (2.02) | 0.00 |
| Quality of Barn Management | 4.48 (1.65) | 5.88 (0.45) | 0.00 |
| Skills in Horsemanship Activities | 9.06 (4.33) | 14.16 (2.07) | 0.00 |
| Skills in Team Building | 4.25 (2.07) | 5.83 (1.55) | 0.0 |
| Total Skills Evaluation Score | 26.92 (9.91) | 39.75 (4.19) | 0.00 |

The equine handling skill evaluation was scored on a 0–56 point scale.

When evaluating each group, there were no significant differences between groups for both overall scores and for each category ($p > 0.05$; Table 7). The one-way ANOVA indicated that LOS ($p = 0.25$) and PL ($p = 0.09$) were not significant indicators of equine handling skills when evaluated separately, nor when LOS and PL were evaluated together ($p = 0.38$). The regression analysis did not show a correlation between emotional safety and equine handling skills when evaluating all participants together (R = 0.25, $R^2 = 0.06$). When evaluating each group individually, a positive correlation between equine handling skills and emotional safety was found in groups participating in four or fewer weeks of therapy (Group A, Group B, and Group C), with the group with the greatest PL (Group C) demonstrating the strongest correlation (Table 5).

**Table 7.** Mean difference (SD) of equine handling skill evaluation scores in regard to length of stay (LOS) and weekly participation level (PL) for young adults in a residential treatment program with weekly psychotherapy incorporating equine interaction sessions.

| Group Category | Abilities in Equine Care | Quality of Barn Management | Skills in Horsemanship Activities | Skills in Team Building | Total Skill Evaluation Scores |
|---|---|---|---|---|---|
| Group A ($n = 11$) | 5.46 (3.78) | 2.00 (2.05) | 6.64 (4.18) | 2.09 (1.76) | 16.18 (9.33) |
| Group B ($n = 4$) | 6.50 (2.08) | 1.75 (1.71) | 4.25 (3.35) | 1.50 (1.92) | 14.00 (8.08) |

**Table 7.** *Cont.*

| Group Category | Abilities in Equine Care | Quality of Barn Management | Skills in Horsemanship Activities | Skills in Team Building | Total Skill Evaluation Scores |
|---|---|---|---|---|---|
| Group C (n = 5) | 6.80 (4.87) | 2.00 (1.41) | 6.20 (6.10) | 2.40 (1.67) | 17.40 (13.63) |
| Group D (n = 12) | 5.17 (4.55) | 1.58 (1.98) | 5.92 (5.21) | 1.75 (1.77) | 14.48 (12.25) |
| Group E (n = 21) | 6.25 (4.39) | 2.10 (1.77) | 6.60 (4.19) | 2.50 (1.57) | 17.45 (10.67) |
| Group F (n = 8) | 6.71 (4.12) | 3.00 (1.00) | 6.43 (4.76) | 3.00 (1.16) | 19.14 (11.38) |

The physical skill evaluation was scored on a 0–56 point scale. Groups were divided according to LOS and PL: Group A (LOS—2–4 weeks; PL—1–2 equine activities/week), Group B (LOS—2–4 weeks; PL—3–4 equine activities/week), Group C (LOS—2–4 weeks; PL—5–7 equine activities/week), Group D (LOS—5–7 weeks; PL—1–2 equine activities/week), Group E (LOS—5–7 weeks; PL—3–4 equine activities/week), Group F (LOS—5–7 weeks; PL—5–7 equine activities/week). No significant differences were found between groups ($p = 0.05$).

## 4. Discussion

Even prior to the COVID-19 pandemic, the use of the horse as a therapy tool in residential substance abuse treatment programs was growing in popularity [22]. This trend has been attributed to the fact that traditional talk therapy had established a pattern of being less effective for the younger population in addiction treatment programs [33]. Experientially based interventions such as PIE provided another opportunity to establish change in the mental health community, particularly as it pertained to young adults [34]. A well-documented benefit to therapeutic equine interaction was a longer engagement in the therapy process for those experiencing substance abuse, which is imperative in early recovery [22,35]. Participation, nonetheless, within therapeutic interventions became hindered with the onset of the COVID-19 pandemic, so that the offering of long-term programs and widespread in-person therapeutic interventions dwindled [6,7]. Further, while participation continues to be a valuable component of the treatment process, the question remained as to whether or not individuals in these addiction treatment programs benefited from the therapeutic equine interaction beyond just improved retention rates [22]. Thus, the objective of this study was to determine the efficacy of PIE as it pertains to the development of emotional safety and learning in young adults participating in a residential addiction treatment program during the COVID-19 pandemic and whether the length of stay or the level of participation had an impact on the effectiveness of this treatment approach.

### 4.1. Impact of Length of Stay and Participation Level

Despite the difficulties faced by residential treatment programs during the COVID-19 pandemic [4,7], young adults in the residential addiction treatment program within the current study improved in their overall scores associated with emotional safety and semantic and procedural memory. This improvement was not dependent on how long patients stayed in the program or how many times per week they participated in the equine interactive activities. In fact, differences between groups were insignificant for emotional safety and procedural memory. Further, for semantic memory as evaluated by the equine knowledge exam, it was the group with the shortest LOS and least PL that reflected the greatest improvement. Additional equine interaction studies have supported positive responses in therapeutic inventions lasting only two weeks [36–38] compared with other nonequine-based therapy programs requiring a longer treatment process to see benefits [39]. This suggests the early impact of equine interactions has a profound effect on engaging the patient in the therapeutic process, as observed previously in improved retention rates [22]. Previous research has reported that even after one minute of canine therapeutic interaction cortisol concentrations dropped within the therapy participant [40]. This effectiveness of

animal interaction in reducing cortisol levels is important to note, as high levels are not only reflective of stress, but also have an impact on cognitive function that can jeopardize learning, with high cortisol levels being associated with hinderance in youth cognitive function [41]. Further, drug addiction has been documented to alter the functioning of the hypothalamic–pituitary–adrenocortical (HPA) axis, which in turn impedes cortisol reactivity, which allows the body to lower cortisol levels when stress is not present [42]. This dysregulation of cortisol during withdrawal is believed to obstruct the ability of the SUD patient to learn sober living habits during therapy [13]. The current findings documenting the ability to learn within a short timeframe suggest potential improvement in the function of the HPA axis and associated cortisol reactivity. This is not only important for the patient to respond accordingly to stressors, but allows for improved cognitive functioning during the therapeutic process. Nevertheless, research concerning cortisol levels within PIE participants is lacking for those going through the withdrawal process, and, as such, further research is warranted to investigate cortisol concentrations in PIE participants within a residential substance abuse treatment program.

Equine interaction within psychotherapy lends itself to a more time-efficient option when working to achieve similar results, which is particularly beneficial to young adults, where engagement in programs can be limited [43]. Further, this is beneficial for mental health professionals that struggled with offering therapeutic interventions during the COVID-19 pandemic [4,6]. In addition, while participants did experience withdrawal during their stay at the treatment center, no participant was removed from PIE sessions due to health-related concerns, including COVID-19. This is important to note, as a direct relationship between poor mental health and the severity of COVID-19 has been documented [44,45], and, thus, a therapeutic treatment option that demonstrates improvement in emotional safety within a shortened timeframe holds promise. This participation level through withdrawal is further promising, as compliance is low with traditional therapeutic interventions during this early stage of the treatment process [9], and yet compliance and treatment success are critical, as the population with SUD is more susceptible to health-related issues, including COVID-19 and the associated adverse outcomes [16,44].

### 4.2. Physical Activity and Mental Health

While all categories assessed in equine-related procedural memory demonstrated improvements, it was only certain areas in the semantic memory that improved within the current study, including that of riding topics. This may be due to the popularity of one of the additional equine interactive activities, which was trail riding. In fact, while all the young adults in this study participated in the weekly PIE session, the most common additional equine interaction as documented by the mental health professionals during the residential stay was the optional trail riding. Residential addiction treatment programs within the United States typically do not have access to trail riding opportunities for their patients [22]. Further, the physical instability clients can experience during the withdrawal process and how that might impact the physical requirements associated with equine interactive activities like trail riding [46] may be a hinderance for facilities to offer such programs. Nevertheless, withdrawal symptoms did not limit participation in these equine interactive activities in the current study, and this may be due to the fact that somatic symptoms associated with withdrawal are less intense for younger adults [47], allowing for the safe implementation of physical activities to an addiction treatment program designed for a younger population. Further, the mere presence of the animal within the therapeutic environment facilitates the release of oxytocin, which is a stress-modulating hormone [48,49]. This increase in oxytocin during animal interaction decreases stress, anxiety, and even pain. This response of the body to the animal in conjunction with a physical activity such as riding is particularly important for mental health, as physical activity releases endorphins helping to further lessen anxiety, stress, and depression; boost self-esteem; and reduce the feeling of pain [50,51]. Moreover, participation within trail riding activities may hold additional benefits, as increased physical activity has been documented to improve car-

diovascular health along with cognitive function [52]. While cognitive function was not directly measured, improvements in equine-related memory measured within the current study suggest the benefit of this form of physical activity on cognitive function. This is particularly important, as this improvement was seen during the COVID-19 pandemic when marked hinderance in learning within young adults was documented due to strains on mental health [53,54]. Interestingly, despite the risks of COVID-19 contraction [5] and the difficulties associated with the withdrawal process [47], trail riding was one of the most popular equine interactive activities for participants, and thus may be a driving force for treatment compliance and for longer stay in a residential program if this option is available. Further research concerning the use of trail riding activities in a residential treatment program compared with other programs not offering these options should be a consideration to objectively determine the benefits.

The improvements in emotional safety seen in the current study were not dependent on a lengthy stay in the residential treatment program, nor were they dependent on a high participation rate in equine interactive activities beyond the weekly PIE sessions. In fact, on the contrary, there was a relationship seen between emotional safety and procedural memory development within those participants only staying for 2–4 weeks. Further, while the relationship became stronger as the level of weekly participation increased, even the minimum participation of just once a week still documented this relationship. On the other hand, for the development of semantic memory, this relationship with emotional safety was only found in one group, those participating in three to four weekly equine interactive activities, and this relationship was weak. This relationship between emotional safety and equine handling skills emphasizes that for those young adults doing a short-term stay of only 2–4 weeks in a residential addiction treatment program, the curriculum may need to focus more on the development of procedural memory than semantic memory, as this may have more of an impact on emotional safety at this point in the recovery process. This improvement is supported by research indicating physical exercise and movement are effective strategies for addiction treatment [55]. Additionally, physical activity allows for a release of unused energy, decreasing problem behaviors and increasing focus during therapy sessions [56]. Federman (2011) [57] points out the benefits of movement within therapy stating, "The body leads human beings to connect to themselves and to their social and physical environment". This kinesthetic learning "aids self-discovery" and "enables to find security" during the therapy process [57], as seen in the improvement in self-esteem and personal security scores within the current study. As such, this development of procedural memory should be a vital component to the PIE curriculum to encourage the development of emotional safety in the early stages of a residential treatment program.

*4.3. Limitations*

This study is of value to the mental health community, as it is the first to quantify the impact of equine interaction on emotional safety and learning for young adults within a residential psychotherapy program for substance abuse. Moreover, the impact of PIE was measured during the COVID-19 pandemic, when substance abuse was on the rise [2,6], helping to address an urgent need within the mental health community. Nonetheless, this study encountered limitations including the lack of participants being tracked past the point of discharge, in which research suggests a positive effect of emotional safety might be demonstrated past the initial treatment protocols [58]. Although this study found a short length of stay and limited weekly participation in equine interactive activities had a positive impact on young adults in these programs, long-term impacts after leaving these programs may be dependent on these variables [35,59]. Further, despite the benefits observed within as little as four weeks, persistent substance abuse has been linked to dysregulation of the HPA axis, leading to a hindered cortisol response to stressors that can persist for four weeks or more after the withdrawal period, making the individual more prone to relapse [42]. As such, while improvements may be observed at the time of the release from the residential treatment program, without tracking cortisol reactivity to

guarantee appropriate response to stressors, relapse cannot be safeguarded. Further, this appropriate response of cortisol to stressors was critical during the COVID-19 pandemic, as young adults faced a multitude of psychological stressors during the pandemic that could have jeopardized abstinence [11]. Thus, future research should include long-term tracking of patients after release from a residential treatment psychotherapy program utilizing PIE to confirm continued abstinence.

## 5. Conclusions

Residential addiction treatment facilities across the nation are integrating equine interaction into their primary treatment curriculum in hopes of engaging young adults who do not benefit from traditional therapy modalities. Equine interaction within residential psychotherapy programs is an experientially based model that allows clients to develop equine-related knowledge and skills through the building of an emotionally safe learning environment. Improvement in emotional safety and semantic and procedural memory was observed within this study by the end of the residential psychotherapy program, and this improvement was not dependent on the length of stay within the program or on participation level in the equine interaction. Further, despite the hurdles associated with the COVID-19 pandemic, these benefits were accomplished with minimal weekly equine interaction during a brief length of stay, within as little as 2–4 weeks, offering support as to the efficacy of this treatment approach for young adults.

**Author Contributions:** Conceptualization, K.H. and M.C.N.; Methodology, K.H., M.C.N., T.P., D.C., B.J.R., P.L.R. and K.G.; Software, K.H.; Validation, K.H., M.C.N., T.P., D.C., B.J.R., P.L.R. and K.G.; Formal Analysis, K.H. and M.C.N.; Investigation, K.H.; Resources, K.H. and M.C.N.; Data Curation, K.H. and M.C.N.; Writing—Original Draft Preparation, K.H., M.C.N., T.P., D.C., B.J.R., P.L.R. and K.G.; Writing—Review and Editing, K.H., M.C.N., T.P., D.C., B.J.R., P.L.R. and K.G.; Visualization, K.H., M.C.N., T.P., D.C., B.J.R., P.L.R. and K.G.; Supervision, K.H., M.C.N. and T.P.; Project Administration, K.H., M.C.N. and T.P.; Funding Acquisition, K.H. All authors have read and agreed to the published version of the manuscript.

**Funding:** This research received no external funding.

**Institutional Review Board Statement:** Review and approval by the Mississippi State University Institutional Review Board (Protocol # 22-482) and Institutional Animal Care and Use Committee (Protocol # 21-306).

**Informed Consent Statement:** Not applicable.

**Data Availability Statement:** Not applicable.

**Acknowledgments:** The authors would like to acknowledge the support from the staff of the American Addiction Centers' Oxford Treatment Center Resolutions Center in Oxford, Mississippi and Dogwood Equine Connection Therapy Center in Starkville, Mississippi.

**Conflicts of Interest:** The authors declare no conflict of interest.

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
