# Peer review of "Psychotherapy Incorporating Equine Interaction as a Complementary Therapeutic Intervention for Young Adults in a Residential Treatment Program during the COVID-19 Pandemic"

_covid, doi:10.3390/covid3100107_

Round 1
Reviewer 1 Report (Previous Reviewer 2)
Paper is well written and can be accepted in current form
Reviewer 2 Report (New Reviewer)
Thank you for the opportunity to review your manuscript which investigates the effectiveness of an alternative treatment for substance use disorder (SUD) among young adults during the Covid-19 pandemic in the USA. Minor additions and corrections are outlined in the attached file. Best wishes for your future research projects.

This manuscript is a resubmission of an earlier submission. The following is a list of the peer review reports and author responses from that submission.
Round 1
Reviewer 1 Report
Abstract has no descriptive statistics, no effect sizes and no inferential tests
Hence ms is not worthy of consideration and should have been rejected at initila stage.
I did not waste my time reading teh MS as abstract was so unsicentific
Author Response
While length of abstract for the journal limits the amount of details that can be given within the abstract, additional details were added to the abstract concerning the statistical analysis and associated results along with further changes made to the abstract as suggested by another reviewer. See attached revised manuscript. Authors hope that these changes may persuade reviewer to assess the full manuscript. Feedback from reviewer is valued in moving forward with current research.

Reviewer 2 Report
1. Do not make abbreviation in abstract.
2. Line 49-50, saying Covid as pandemic but no citational proof
provided. So for this authors can refer: (i) https://doi.org/10.18280/ts.390548
3. Line no. 82- How author can say one non-traditional therapy option
found within residential pro- rams that provided an environment to meet these
challenges associated with the COVID- pandemic was psychotherapy
incorporating equine interaction?
4. Line 139: authors have provide any government agency report to
elaborate- During the participant’s stay at the residential facility,
health status was monitored by the physicians of the treatment center.
5. How the Groups A and C and formed and how they are differ?
6. Provide detailed statistical analysis would enhance the rigor and
reliability of the findings.
7. It is advisable for the authors to include more recent references
that have undergone scientific review on the topic. Additionally, more
references should be added to support the claims made throughout the paper.
8. The advantages and disadvantages of the proposed work should be
highlighted in a more effective manner. Make a separate section for this.
9. Table 7 not cited textually. Check for others also.
10. Line 442- what is buffer against negative feelings”
11. Discussion section is very well written but the presentation is very dull.
Present it with some heading/numbering/bullets etc.
12. Conclusion must conclude the details of the work presented in the
manuscript. Revise it.
13. Refer some latest literatures of 2022, 2023 like (i) https://link.springer.com/article/10.1007/s12652-021-03306-6
14. Paying meticulous attention to the overall writing is crucial, as it
significantly affects the quality of the paper. Ensuring clarity, coherence,
and adherence to proper grammar and style will enhance the overall
readability and professionalism of the manuscript.
Author Response
The following includes the suggested revisions by the reviewer and the revisions that were made by the authors. Updated manuscript is attached. Authors thank the reviewer for their suggestion revisions.
- Do not make abbreviation in abstract.
Abbreviations were removed from abstract, except for the use of “COVID-19” due to the title of the journal and the repetitive use of this abbreviation within other published manuscripts within the journal.
2. Line 49-50, saying Covid as pandemic but no citational proof
provided. So for this authors can refer: (i) https://doi.org/10.18280/ts.390548
Citational proof concerning COVID-19 being referred to as a pandemic within the manuscript was added utilizing journal given by reviewer.
Line no. 82- How author can say one non-traditional therapy option
found within residential pro- rams that provided an environment to meet these
challenges associated with the COVID- pandemic was psychotherapy
incorporating equine interaction?
Section containing line 82 was not specifically discussing residential programs utilizing psychotherapy incorporating equine interaction. Authors apologize for the unclear statements made within this section. As such, the section was rephrased to clarify points being made concerning general use of residential programs, not specific to those using psychotherapy incorporating equine interaction.
Line 139: authors have provide any government agency report to
elaborate- During the participant’s stay at the residential facility,
health status was monitored by the physicians of the treatment center.
Additional information was provided concerning agencies involved with accreditation of the program and reporting guidelines associated with the treatment activities for patients involved with this study.
How the Groups A and C and formed and how they are differ?
Details were given in the methods section under statistical analysis, however, to further explain grouping, this information was moved to the section labeled as “additional equine interaction” with additional details provided to hopefully clarify formation of groups and differences between groups.
Provide detailed statistical analysis would enhance the rigor and
reliability of the findings.
Additional information within the methods section was provided concerning statistical analysis and the section was rewritten.
It is advisable for the authors to include more recent references
that have undergone scientific review on the topic. Additionally, more
references should be added to support the claims made throughout the paper.
Additional scientific references were added throughout the paper to support claims and these additions were more recent references from peer-reviewed scientific journals.
The advantages and disadvantages of the proposed work should be
highlighted in a more effective manner. Make a separate section for this.
Specifics to advantages and disadvantages were added to both the introduction under section 1.3 and within the discussion under section 4.3.
Table 7 not cited textually. Check for others also.
Authors thank the reviewer for noting the missed citing of Table 7. That has been added and the other tables were checked to ensure that they were cited textually within the manuscript.
Line 442- what is buffer against negative feelings”
Sentence was reworded to explain more objectively the impact of the presence of the horse.
Discussion section is very well written but the presentation is very dull.
Present it with some heading/numbering/bullets etc.
Headings were added to the discussion section and numbering was included throughout the manuscript. Additional information was provided included additional citations.
Conclusion must conclude the details of the work presented in the
manuscript. Revise it.
Further information concerning specific results reported within the study were added to the conclusions.
Refer some latest literatures of 2022, 2023 like (i) https://link.springer.com/article/10.1007/s12652-021-03306-6
More current research was added concerning equine interaction and COVID-19.
Paying meticulous attention to the overall writing is crucial, as it significantly affects the quality of the paper. Ensuring clarity, coherence, and adherence to proper grammar and style will enhance the overall readability and professionalism of the manuscript.
Revisions were made throughout the manuscript and additional reviewers were invited to review updated manuscript to ensure content was clear, coherent, and followed proper grammar and scientific style according to journal guidelines.
